# Herbivory induced methylation changes in the Lombardy poplar: A comparison of results obtained by epiGBS and WGBS

A. Niloya Troyee[1][☯]*, Cristian Peña-Ponton[2][☯], Mónica Medrano[1], Koen J. F. Verhoeven[2], Conchita Alonso[1]*

**1** Estación Biológica de Doñana, Consejo Superior de Investigaciones Científicas (CSIC), Sevilla, Spain,
**2** Department of Terrestrial Ecology, Netherlands Institute of Ecology (NIOO-KNAW), Wageningen, The Netherlands

☯ These authors contributed equally to this work.
* niloyatroyee@gmail.com (ANT); conalo@ebd.csic.es (CA)

## Abstract

DNA cytosine methylation is an epigenetic mechanism involved in regulation of plant responses to biotic and abiotic stress and its ability to change can vary with the sequence context in which a cytosine appears (CpG, CHG, CHH, where H = Adenine, Thymine, Cytosine). Quantification of DNA methylation in model plant species is frequently addressed by Whole Genome Bisulfite Sequencing (WGBS), which requires a good-quality reference genome. Reduced Representation Bisulfite Sequencing (RRBS) is a cost-effective potential alternative for ecological research with limited genomic resources and large experimental designs. In this study, we provide for the first time a comprehensive comparison between the outputs of RRBS and WGBS to characterize DNA methylation changes in response to a given environmental factor. In particular, we used epiGBS (recently optimized RRBS) and WGBS to assess global and sequence-specific differential methylation after insect and artificial herbivory in clones of *Populus nigra* cv. 'italica'. We found that, after any of the two herbivory treatments, global methylation percentage increased in CHH, and the shift was detected as statistically significant only by epiGBS. As regards to loci-specific differential methylation induced by herbivory (cytosines in epiGBS and regions in WGBS), both techniques indicated the specificity of the response elicited by insect and artificial herbivory, together with higher frequency of hypo-methylation in CpG and hyper-methylation in CHH. Methylation changes were mainly found in gene bodies and intergenic regions when present at CpG and CHG and in transposable elements and intergenic regions at CHH context. Thus, epiGBS succeeded to characterize global, genome-wide methylation changes in response to herbivory in the Lombardy poplar. Our results support that epiGBS could be particularly useful in large experimental designs aimed to explore epigenetic changes of non-model plant species in response to multiple environmental factors.

**Data Availability Statement:** All the adapter trimmed sequencing data for WGBS and epiGBS libraries have been deposited in ENA, project PRJEB51853 (https://www.ebi.ac.uk/ena/browser/

view/PRJEB51853). The Populus nigra cv. 'italica' reference genome used is available at ENA project PRJEB44889 (https://www.ebi.ac.uk/ena/browser/view/PRJEB44889).

**Funding:** This work was supported and funded by the European Union's Horizon 2020 research and innovation program via the Marie Sklodowska-Curie ITN 'EpiDiverse' (grant agreement No 764965) <https://cordis.europa.eu/project/id/764965> and CSIC Open Access Publication Support Initiative (Unit of Information Resources for Research). C.A. and M.M. were also supported by the Spanish Government through the Research Project 'Epinter' (PID2019-104365GB-I00, Ref. AEI/10.13039/501100011033) <https://www.ciencia.gob.es/> The funders had no role in study design, data collection and analysis, decision to publish, or preparation of the manuscript.

**Competing interests:** The authors have declared that no competing interests exist.

## Introduction

DNA methylation is an epigenetic modification found in the genomes of most living organisms, from bacteria to plants, animals and fungi [1, 2]. Cytosine methylation is the most common form of DNA methylation and it indicates the addition of a methyl group (CH3) at 5'-carbon pyrimidine ring of cytosine nucleotide [2], hereafter referred to as DNA methylation. The frequency and location of DNA methylation vary drastically among animals and plants. In animals, it predominantly occurs on guanine- and cytosine-rich regions of the genome, specifically in CpG context, while in plants, DNA methylation is more extensive and can also be found in CHG and CHH (H = Adenine, Thymine, Cytosine) contexts, catalyzed by a diverse set of methyltransferase enzymes [2, 3]. Plants exhibit widespread variation in DNA methylation among and within plant families. Specifically, the global DNA methylation level in angiosperms has been linked to variance in genome size [3, 4] and to some extent to other intrinsic and extrinsic factors such as life history traits and regional provenance [5, 6]. Regarding the relative frequency of DNA methylation in different sequence contexts, methylation is ubiquitous and relatively frequent in CpG context for the majority of plant species, whereas methylation levels in CHG are reduced in species of the Brassicaceae family, and tend to be low for CHH in the Poaceae family [3]. As regards the location within genomes, methylation in the three contexts is highly enriched in repetitive DNA, transposons, and pericentromeric regions, whereas genic methylation tends to occur mainly in the CpG context [7, 8].

At the intraspecific level, variation in DNA methylation has been linked to geographic and environmental gradients in wild plant populations (see e.g., [9–12]). Furthermore, sound evidence for changes in DNA methylation associated to abiotic stress [9, 13] and biotic interactions have been predominantly recorded in model or crop plants (reviewed in [14]). In particular, DNA methylation variation has been associated with plant defense against herbivores in several species [15–20]. For example, in *Brassica rapa*, foliar herbivory is associated with both DNA demethylation and changes in pollinator-relevant floral traits that decrease the attractiveness of the plants to their main pollinators [20]. Most of the referred studies above have used either global estimates of methylation or anonymous markers to detect overall changes in species lacking a reference genome, impeding the ability to infer the genomic location and potential function of observed methylation changes in response to insect or artificial herbivory (but see for instance [21, 22]).

Whole Genome Bisulfite Sequencing (WGBS) and Reduced Representation Bisulfite Sequencing (RRBS) are two methods for studying DNA methylation, with RRBS being a cost-effective method for quantifying methylation that targets a small and restriction enzyme site-associated fraction of the genome [23]. Bisulfite sequencing includes pre-treating DNA with sodium bisulfite, which converts cytosine to uracil while 5-methylcytosine remains unchanged, followed by sequencing and data processing to identify methylation at single cytosine site resolution [24, 25]. WGBS can cover entire genome and is regarded as the gold standard for studying DNA methylation because its single-base resolution provides information about the sequence context and offers the possibility for mapping to entire genomes. The quality of the output will vary with genome features (e.g., genome size, frequency of repeats), sequencing depth reached and the quality of the annotated reference genome, which limits its application to non-model plants with unknown genome features [23]. RRBS, in contrast, targets a reduced fraction of the genome using restricted digestion that allows studying species with varying genome sizes and species that lack a reference genome at a lower cost. Among the various RRBS techniques available, epiGBS [23, 26] is a method with specific barcoded adapters that uses genotyping by sequencing of bisulfite-converted DNA. Bioinformatic processing of obtained sequencing data provides a reliable de novo reference for the genomic loci that are

targeted by the method, or uses an existing reference genome if available, for inferring both methylation quantification and single nucleotide polymorphism detection [26, 27]. EpiGBS has been successfully used to estimate overall DNA methylation variation in a variety of non-model species, including mosses, snails, birds, and plants [28–32]. Furthermore, estimation of DNA methylation by epiGBS correlates well with estimates obtained by WGBS for several accessions of the model species *Arabidopsis thaliana* [27]. Nonetheless, since epiGBS inherently captures only a certain fraction of the genome, it is unknown how effective the method is at capturing the methylation response to specific stress factors [23]. Therefore, to move forward in analyzing the epigenetic contribution to plant responses to stress, it is crucial to corroborate how similar are the outputs of epiGBS and WGBS to comprehend specific stress responses in non-model plants. If the two techniques identify similar global and context-specific methylation shifts and point to similar genomic location of most of the observed methylation changes in response to a certain level of environmental stress, the epiGBS analyses could be useful to explore the links between epigenetic variation and plant functional phenotypic traits in non-model plants, with typical ecological designs involving large sample sizes and multiple levels of environmental variation that have mainly used anonymous markers [15, 16, 19] or indirect evidence of epigenetic contribution to stress response [13].

In this study, we used both WGBS and epiGBS techniques to evaluate changes in DNA methylation in cuttings of the Lombardy poplar, *Populus nigra* cv. 'italica', in response to insect herbivory and artificial herbivory. Lombardy poplars are fast-growing trees that have a clonal origin and widespread distribution, which makes this cultivated variety an excellent study system to investigate epigenetic responses to specific factors with reduced variation at the genetic level. Furthermore, in long-lived plants, rapid and reversible methylation changes can contribute to plant phenotypic plasticity [33–35] and could perhaps be associated with transcription changes observed after insect and artificial herbivory in poplar [36]. Specifically, here we addressed the following questions: i. To what extent genome-wide methylation changes induced by herbivory can be similarly detected using epiGBS and WGBS outputs? ii. How frequent and strong are these methylation changes in the three sequence contexts according to the two techniques? iii. Are there technical/biological biases between the two techniques with regard to the frequency of herbivory-induced methylation variations in specific genomic sites (gene body, gene promoter, transposable element, etc.)? iv. Can the two techniques identify functional changes associated with the plant methylation response to herbivory?

## Methods

### Study system

**Plant species.** Poplar is a suitable genus for genome-wide investigations due to its compact genome size (~500 Mb) and availability of reference genomes [37]. Black poplar, *Populus nigra* L. (Salicaceae), is a diploid deciduous tree native to north-west Europe that grows in floodplain woods and riparian environments [38]. We used the clonal Lombardy cultivar (*Populus nigra* cv. 'italica' Duroi) because it can be easily propagated by cuttings and has low genetic variation [39]. We did not require any special permit for sample collection because our study plants came from the Marburg Botanical Garden (Germany).

**Herbivory types.** We used two forms of tissue damage: true insect herbivory and simulated artificial herbivory to decipher the changes induced by herbivory in terms of DNA methylation. Larvae of the gypsy moth, *Lymantria dispar* (L.) (Erebidae), were used for insect herbivory treatment. This polyphagous insect is a major pest of northern hemisphere hardwood forests, fruits, and ornamentals, that has over 500 host plant species including poplars [40], it has a short generation time with precisely defined developmental stages and larvae are

easy to manipulate. We obtained L2 instar larvae from Dr. Sybille Unsicker's lab (Max Planck Institute for Chemical Ecology, Jena, Germany) and kept them in a climate chamber (14/10 h light/dark, 20 ºC, 60% humidity) feeding on artificial diet (MP Biomedicals LLC) until the experiment was conducted. Two-three days before the start of the herbivory treatment, larvae were fed poplar leaves to get them adapted to this food source. For the insect treatment, larvae were always placed on fully expanded leaves (see below for further details). The artificial herbivory treatment was performed by mechanically punching holes in the leaves and spraying them with a solution of Jasmonic Acid (JA), which is known to be an important chemical elicitor of many plant defense responses [41]. These two complementary treatments were selected because *Populus* species develop secondary defense chemicals in response to jasmonates [36]. Also, when black poplar is attacked by gypsy moth caterpillars, it produces a variety of direct and indirect defenses, including an increase in JA on damaged leaves and several volatile organic compounds that attract herbivore enemies [40, 42].

## Experimental design

**Plant materials and growth conditions.** Lombardy poplar clones were generated through vegetative propagation by rooted cuttings from three adult parent trees located in Italy and grown in a common garden in the Marburg Botanical Garden (Germany) for 10 months [39]. Each garden planted tree from which all the members of a clone have descended will be denoted hereafter as an ortet. Cuttings of approximately 30 cm in length were sampled from the common garden trees and stored at 4 ºC and dark conditions for 2 weeks. Cuttings were soaked (2 cm bottom) in a rooting solution (50 mg/L Rhizopon AA 50 mg tablets) overnight and planted in 2 L pots (3 cuttings per pot) with 1:1 sand:peat mixture (30% coarse sand, 20% fine sand, and 50% nutrient-poor potting soil) and 5 ml of rooting solution. Pots were maintained in a flood table with regular watering to pot capacity for 2 weeks and then rooted cuttings were transplanted to individual 2 L pots with the same 1:1 sand:peat mixture and watering regime. Cuttings were fertilized with slow-release fertilizer Osmocote Exact mini (15-9-11+2MgO+TE) after two weeks (2 g) and 10 weeks (1 g) of transplanting. Greenhouse conditions during the experiment included: temperature: (day/night) 22/18 ˚C (±2˚C), relative humidity: 60% (±5%), light: (day/night) 16/8 h.

A total of 27 similar-sized and 15-week-old cuttings (ramets hereafter) originated from the three ortets (i.e., nine ramets per ortet) were used in the experiment. Ramet is defined as an individual obtained clonally from an ortet, and thus, the nine ramets derived from a certain ortet should be genetically identical individuals. Three ramets per ortet were included in each of the three experimental treatments: control, insect herbivory, and artificial herbivory. We did not assort the plants from different treatments together in order to avoid volatile organic compounds exchange [40]. Per treatment, the ramets were randomly distributed in two trays (flood tables) and were randomized regularly.

**Herbivory treatments procedure.** To control for potential positional effects, damage was always inflicted on leaves of the basal half of the main branch of each ramet and methylation changes will be determined from material taken from the most adjacent undamaged leaves grown in the apical half of that branch. Both insect and artificial herbivory treatments were repeated twice to enable a priming effect to finally get a stronger and/or faster response [17, 43]. For priming the plants in insect herbivory treatment, ten *L. dispar* L2 instar larvae were placed on full expanded leaves of the main branch of our experimental poplar ramets, which were enclosed using nylon mesh bags (75*100 cm). After five days, the larvae were removed for three days so the plants could recover. For the second herbivory induction event, seven L2 instar and five L4 instar larvae were placed on leaves of the lower part of the main branch,

enclosed within a nylon mesh bag (75*100 cm) and allowed to feed freely for sevendays before the collection of leaf samples (see below).

Artificial herbivory was conducted in the main branch and in a similar location as insect herbivory. In the priming phase, 6–8 holes per leaf (ca. 3–5 mm diameter) were punched in three to four leaves of the basal half of the main branch. Immediately after the artificial wounding of the leaves, two pumps (150 μL/pump) of a JA (Sigma J2500- 100MG) solution were sprayed on the damaged leaf and three pumps all over the plant, repeating this procedure twice. During the second herbivory event similar number of holes as for priming were punched in each of 10–12 leaves and JA solution was sprayed four-times and enclosed within nylon mesh bags as described above. The JA was solubilized in ethanol and diluted in deionized water to a 1-mM JA solution with 0.1% Triton-x 100 as a surfactant to increase cuticle penetration [36].

In the control group, similar-positioned, well-developed leaves from the main branch of each ramet were sprayed with an equivalent aqueous solution in which no JA was added and covered with nylon mesh bags in a similar manner as the herbivore treated ones. The experiment finished 17 weeks after clonal propagation.

## Laboratory methods and library preparation

**Sampling and DNA extraction.** We collected tissue from undamaged and fully expanded leaves of the adjacent apical half of each ramet, either 24 hours after the second herbivory event in treated plants or after the aqueous spraying in controls. We kept these leaves without any bag cover throughout the duration of the experiment. A cork borer was used to take 5–6 discs of leaf tissue (ca. 3–5 mm diameter), that were stored in labelled vials and immediately frozen in liquid nitrogen. Vials were kept at -80 ˚C until DNA extraction. Sampling and DNA extraction order were determined by randomly selecting one sample per treatment (irrespective of the ortet) at a time. Frozen leaf material was disrupted and homogenized using a Qiagen TissueLyser II with two stainless steel beads (45 seconds at a frequency of 30.00 1/s). We used Macherey-Nagel NucleoSpin Plant II kit to do DNA extraction and cell lysis Buffer PL1 (CTAB method) to get optimum DNA quality. For each sample, an aliquot of 35 μl at 30 ng/μl was used for epiGBS and at least 1 μg of DNA from a stock solution at minimum 20 ng/μl was used for WGBS. It is important to remark that this is the first time that plant DNA from identical plant individuals has been analyzed using these two techniques.

**epiGBS library construction, sequencing and pre-processing.** We followed the epiGBS2 protocol [27] with few modifications. In brief, samples were randomized and DNA digested using restriction enzymes AseI and NsiI. Hemi-methylated adapter pairs, containing barcodes of sample-specific 4–6 nucleotides, were then ligated to the digested DNA. The barcodes were followed by three random nucleotides and an unmethylated cytosine used to estimate the bisulfite conversion rate. Next, the samples were multiplexed, concentrated and smaller fragments (<60 bp) were removed by NucleoSpin Gel and PCR cleanup Kit (Macherey-Nagel™). SPRIselect magnetic beads were used to select DNA fragments of 300 bp (and lower). Deoxynucleoside triphosphates (dNTPs) that contain 5-methylcytosine were used to repair the nicks produced by hemi-methylated adapters to obtain completely ligated and methylated adapters. We used the EZ DNA Methylation-Lightening kits protocol for converting the multiplexed samples. PCR-amplification of converted DNA was done followed by a final clean-up and size selection. The obtained library was then sequenced paired-end (PE 2x150bp) in one lane of an Illumina HiSeq 4000 sequencer with a 12% phiX spike.

Both the 'reference' branch and 'de novo' branch of epiGBS2 pipeline were used to analyze sequencing data [27]. All the steps were embedded in a Snakemake (version 6.1.1) workflow

[44]. Firstly, removal of PCR duplicates was performed based on the inserted 3-random nucleotide sequence in the adapter sequences. This step was done to confirm true PCR clones so it can be removed from the sequencing data but not the biological duplicates. Then Stacks 2 software [45] was used to demultiplex the samples in accordance with the barcodes followed by adapter trimming using Cutadapt (data stored in European Nucleotide Archive (ENA) project: PRJEB51853) [46]. The bisulfite conversion rate, estimated based on the number of correctly bisulfite-converted control cytosines within the adapters (see [27] and above), was found globally satisfactory ($\geq$ 94.88%). The pipeline then maps the sequence fragments of experimental data with the reference or consensus genome using Bismark v0.19.0 [47] with the default settings. In the epiGBS2 reference branch (epiGBS-R hereafter), the *P. nigra* cv 'italica' reference genome (available at ENA project: PRJEB44889) was used for mapping with default parameters (sequence identity in the last clustering step). In epiGBS *de novo* branch (epiGBS-D hereafter), the genome generated from consensus clusters was used to map the fragments. The final output was a Bismark report file for each sample that contains lines with chromosome/scaffold name, genomic position, strand information, methylated cytosine number, unmethylated cytosine number, cytosine sequence context name (CG/CHG/CHH) and true trinucleotide context information.

In the end, for the epiGBS library, 240 million (243,507,236) reads were successfully demultiplexed and assigned to individual samples (N = 27 ramets). In epiGBS-D, a *de novo* assembly produced 106,267 clusters of 32–290 bp in length (mean = 224 bp), with an average of 1.4 fragments per contiguous cluster (minimum = 1 and maximum = 437).

**WGBS library construction, sequencing and pre-processing.** Preparation of DNA libraries for bisulfite sequencing was performed by IGA Technology Services (Italy) using the Ovation Ultralow Methyl-Seq System (NuGEN, Redwood City, CA) following the manufacturer's instructions. Library preparation order followed the same randomized sample design as for DNA extraction. Libraries were sequenced paired-end (PE 2x150bp) targeting 25X coverage on an Illumina NovaSeq6000 sequencing system. Libraries were randomized and sequenced in two lanes. The sodium bisulfite non-conversion rate was calculated as the percentage of cytosines sequenced at cytosine reference positions in the chloroplast genome and it was found globally satisfactory with conversion $\geq$ 96.37%.

Sequenced reads were processed using the EpiDiverse Toolkit (WGBS pipeline v1.0, https://github.com/EpiDiverse/wgbs) [48]. Briefly, low-quality read-ends were trimmed (minimum base quality: 20), sequencing adapters removed (minimum overlap: 5 bp), and short reads (<36 bp) discarded. The remaining high-quality reads were aligned to the *P. nigra* cv. 'italica' reference genome mentioned above using erne-bs5 alignment package (http://erne.sourceforge.net) allowing for 600-bp maximum insert size, 0.05 mismatches, and unique mapping. Per-cytosine methylation metrics were extracted using MethylDackel (https://github.com/dpryan79/MethylDackel). Three bedGraph files per sample were produced, corresponding to cytosines on each sequence context: CpG, CHG and CHH. Each of these files contained a matrix where each line consists of six columns that indicated the scaffold name, start coordinate, end coordinate, methylation percentage, number of alignments reported methylated bases and unmethylated bases. For the WGBS library, after adaptor trimming and quality control, 200 million (200,660,447) reads per sample were processed, and a total of 52 million (52,885,999) reads were demultiplexed, assigned to individual samples and mapped for further analysis.

**epiGBS and WGBS data filtering.** The downstream analyses required to estimate changes in methylation status between study groups were conducted with methylKit [49] based on Bismark report files. Bismark file of every sample was checked for global low read coverage based on the retained positions and approved since more than 50% of total positions

**Table 1. Total number of cytosines captured by epiGBS-R, epiGBS-D and WGBS techniques in the three sequence contexts (CpG, CHG and CHH) according to the sample representation threshold applied.**

| | | Sample representation | | |
|---|---|---|---|---|
| | | 2/3 samples | all samples | |
| Technique | Sequence context | No. cytosines* | No. cytosines* | % |
| epiGBS-R | CpG | 167,193 | 24,126 | 14.43 |
| | CHG | 260,884 | 36,033 | 13.81 |
| | CHH | 1,394,947 | 172,631 | 12.37 |
| epiGBS-D | CpG | 108,462 | 62,456 | 57.58 |
| | CHG | 162,351 | 92,952 | 57.25 |
| | CHH | 877,942 | 457,165 | 52.07 |
| WGBS | CpG | 10,530,726 | 210,014 | 1.99 |
| | CHG | 16,019,281 | 330,541 | 2.06 |
| | CHH | 90,235,706 | 963,809 | 1.07 |

* Values indicate the total number of cytosines included in each dataset after doing the minimum read coverage filtering ($\geq$ 6 sequencing coverage in WGBS; $\geq$ 10 in epiGBS-R and epiGBS-D), taking into account their presence in at least 2/3 of study samples per group or being common to all study samples.

had sufficient read depth in all cases. Next, cytosine loci with five or fewer sequencing read depths for WGBS (6x) and nine or fewer reads for epiGBS-D and epiGBS-R (10x) were removed and the retained data stored as flat file databases. We applied different minimum read coverage because data from the two sequenced libraries indicated that epiGBS got higher coverage than WGBS for the captured cytosines (S1 Fig). Individual databases were later merged using the *unite* function of methylKit that kept bases covered by 2/3 of the samples per treatment group (i.e., in six out of nine samples). We retained a total of 1,823,024 (epiGBS-R), 1,148,755 (epiGBS-D) and 116,785,713 (WGBS) cytosines. Furthermore, in order to compare only positions with methylation calls that were common to all samples, a dataset was built without any missing values (i.e., data available in 100% of samples). The number of cytosine positions in the three sequence contexts captured by each technique after read coverage filtering and the two sample representation options are shown in Table 1.

## Data analyses

**Methylation levels in epiGBS and WGBS.** Estimates of methylation levels in the three sequence contexts (CpG, CHG, CHH) and genomic features (promoters, gene bodies, downstream, intergenic region) taking into account whether they overlap or not with transposable elements, were calculated for positions that were common to all samples using epiGBS-R, epiGBS-D and WGBS data. Methylation level (%) of a particular site was calculated as: (methylated cytosine read count)/(methylated cytosine read count + unmethylated cytosine read count) * 100. The average methylation was subsequently estimated as the mean of all sequenced cytosines in a sample. Average methylation percentage in each context was compared among epiGBS-D, epiGBS-R and WGBS by Pearson's Chi-square test using *chisq.test* function of stats R package (v3.6.2). Divergence in average methylation (%) between study samples was evaluated independently for each technique with a linear model including herbivory (with three levels) and ortet (with three levels) as fixed factors. Significance of fixed factors and their interaction was tested using the function ANOVA (package car, v3.0.12) [50]. Additionally, we have conducted a technical analysis with the common fragments obtained by both epiGBS-R and WGBS, i.e. using only the subgroup of fragments that had identical coordinates when mapped to the reference genome (average fragment size of 204 bp). But since the results

of this technical comparison are very specific and does not change substantially our main conclusions, to improve readability and conciseness, they are provided as a supplementary material (S1 Appendix).

**Differentially methylated cytosines (DMCs).** In order to detect cytosine loci with a significant shift in methylation, monomorphic positions in our dataset, i.e. those that were always unmethylated or fully methylated were removed. In addition, we removed all positions where the sum of methylated reads across all samples was less than 10 (i.e., very low methylated positions) from the merged file (see epiGBS and WGBS data filtering section in Methods). The filtered dataset was analyzed with generalized linear models as implemented in the R package methylKit [49], that assumes that the methylated to unmethylated counts follow a binomial distribution and the effect of the fixed factors can be estimated with a log-likelihood test for logistic regression. methylKit allows parameter adjustment to identify DMCs corrected by multiple testing based on $q$-value ($q$-value $<$0.05), the minimum methylation difference (fixed = 10%), and direction of methylation shift (hyper or hypo). DMCs were called separately for insect and artificial herbivory treated plants in comparison with control plants, each model included herbivory (control $vs$ treated) as fixed factor and ortet as covariate. Finally, we searched for "stress-specific" DMCs that were present only in one type of herbivory and "non-specific" DMCs that were common to both herbivory treatments. The same criteria and statistical model were applied to epiGBS-D, epiGBS-R and WGBS filtered data if not stated otherwise. Furthermore, Pearson's Chi-square test ($\alpha$ = 0.05) was used to test similarity in number of DMCs obtained between two methods (*chisq.test* funtion stats R package v3.6.2).

**Differentially methylated regions (DMRs).** In WGBS analyses is frequent to adopt a regional approach that considers the non-independence of DMCs that are close to each other within the genome, and combines them into regions (DMRs) to study DNA methylation differences between groups of samples [51]. We call DMRs based on the following procedure: BedGraph files from EpiDiverse/WGBS pipeline were used as input for differential methylation analysis using the EpiDiverse/DMR pipeline v1.0 (https://github.com/EpiDiverse/dmr), each treatment group was compared to the control group and the three cytosine sequence contexts were analyzed separately. Briefly, DMRs were identified by metilene https://www.bioinf.uni-leipzig.de/Software/metilene) [52], using the following parameters: minimum read depth per position: 6; minimum cytosine number per DMR: 10; minimum distance between two different DMRs: 146 bp; per-group minimal non-missing data for estimating missing values: 0.8; adjusted p-value (Benjamini-Hochberg) to detect significant DMRs: 0.05. Only significant DMRs with methylation difference $>$10% between control and treated group were retained for analysis. Due to the short nature of epiGBS fragments (average length 207 nucleotides in the filtered data set), no formal DMR tests were performed.

**Structural annotation of DMCs and DMRs.** We overlapped DMCs and DMRs from the former analyses with *P. nigra* cv. 'italica' genome (ENA project: PRJEB44889) to identify genomic features with differential methylation. We defined three genomic features: gene body was defined as the entire gene from the transcription start site and to termination transcription site, the promoter was defined as the region $<$ 2 kb up-stream of transcriptional start site, and the region located $<$ 2 kb down-stream of termination site was named as downstream. DMCs or DMRs located out of those three features were assigned to intergenic. Coordinates of DMC or DMRs were used to perform the BEDTools intersect command and a custom script for annotating each genomic feature including Transposable Elements (TE). Visualization of the distribution of DMCs for the treatment groups was carried out using custom R scripts (R Development Core Team, 2020).

**Functional analysis of genes associated to herbivory-induced DMRs.** Each DMR was associated with its overlapping gene and/or with the closest gene (maximum 2kb upstream

from transcription start site). Genes associated with either insect or artificial herbivory DMRs were subjected to gene ontology (GO) enrichment analysis. The gene set background (universe) was built with the closest *A. thaliana* homologue of each *P. nigra* cv 'italica' gene, which was determined using BLAST best reciprocal hits of the protein sequences (R package ortho-logr). Best hits were filtered by keeping alignments covering at least 60% of both *A. thaliana* and *P. nigra* proteins, and minimum 60% similarity. *Arabidopsis* sequence proteins were extracted from phytozome V13, and functional annotations were retrieved from the PLAZA 5.0 dicots database (https://bioinformatics.psb.ugent.be/plaza/). GO enrichments were performed using clusterProfiler v4 [53]. P-values were adjusted for multiple testing controlling the positive false discovery rate (q-value).

## Results

### Methylation levels estimated by epiGBS and WGBS

As regards to methylation percentage recorded per cytosine, we found consistent outputs between the three techniques, with cytosines in CpG and CHG contexts showing a bimodal distribution with a much higher frequency in methylation levels < 25%, while cytosines in CHH context showed unimodal distribution skewed towards low methylation and almost no case with methylation levels > 50% (Fig 1). The average cytosine methylation level among all the poplar samples studied (N = 27) ranged from 8.1% to 28.2% according epiGBS-R, from 7.5% to 35.2% according to epiGBS-D, and from 8.0% to 38.79% according to WGBS.

Methylation percentage was higher in CpG, intermediate in CHG, and lower in CHH, regardless of the technique used for estimation. In epiGBS-R, methylation percentage across all samples averaged 27.8 ± 0.1%, 15.6 ± 0.1%, and 8.4 ± 0.1% for CpG, CHG, and CHH contexts, respectively. In epiGBS-D, average methylation obtained were similar to those in reference branch: 32.4 ± 0.1%, 19.0 ± 0.1%, and 5.7 ± 0.1% for CpG, CHG, and CHH context, respectively. In WGBS, average methylation percentage was 38.0 ± 0.5%, 34.9 ± 0.5%, and 9.5 ± 0.8% for CpG, CHG, and CHH contexts, respectively. The chi-squared tests conducted for each context indicated that average DNA methylation per context (i.e., interpreted as a relative proportion) was not statistically different among the three methods for CpG and CHH (Pearson's Chi-square test, CpG $X^2$ = 1.61, df = 2, $P$ = 0.45 and CHH: $X^2$ = 0.99, df = 2, $P$ = 0.60) but significantly different for the CHG context ($X^2$ = 9.12, df = 2, $P$ = 0.01).

For each context, the linear model and ANOVA test applied to evaluate the effect of herbivory treatment and ortet indicated that average methylation percentage was significantly different among the three ortets for CpG and CHG according to epiGBS-R and epiGBS-D (Table 2). In CHH context, average methylation percentage was significantly different among levels of the herbivory treatment according to epiGBS-R and epiGBS-D (Table 2), methylation being lower in control plants (Fig 2). In WGBS, however, neither the ortet or treatment had a significant effect on average DNA methylation recorded in any sequence contexts (Table 2). Given the similarities between epiGBS-D and epiGBS-R, only comparisons between WGBS and epiGBS-R are discussed further.

### Methylation changes induced by herbivory detected as DMCs and DMRs in epiGBS-R and WGBS

In total, epiGBS-R captured 10,675 DMCs, whereas WGBS was able to detect only 4,746 DMCs (S1 Table) for the two herbivory treatments combined. Such a difference between the two techniques was mainly due to a lack of captured DMCs at CHH context when analyzing WGBS data, likely due to reduced statistical power associated to multiple-testing correction

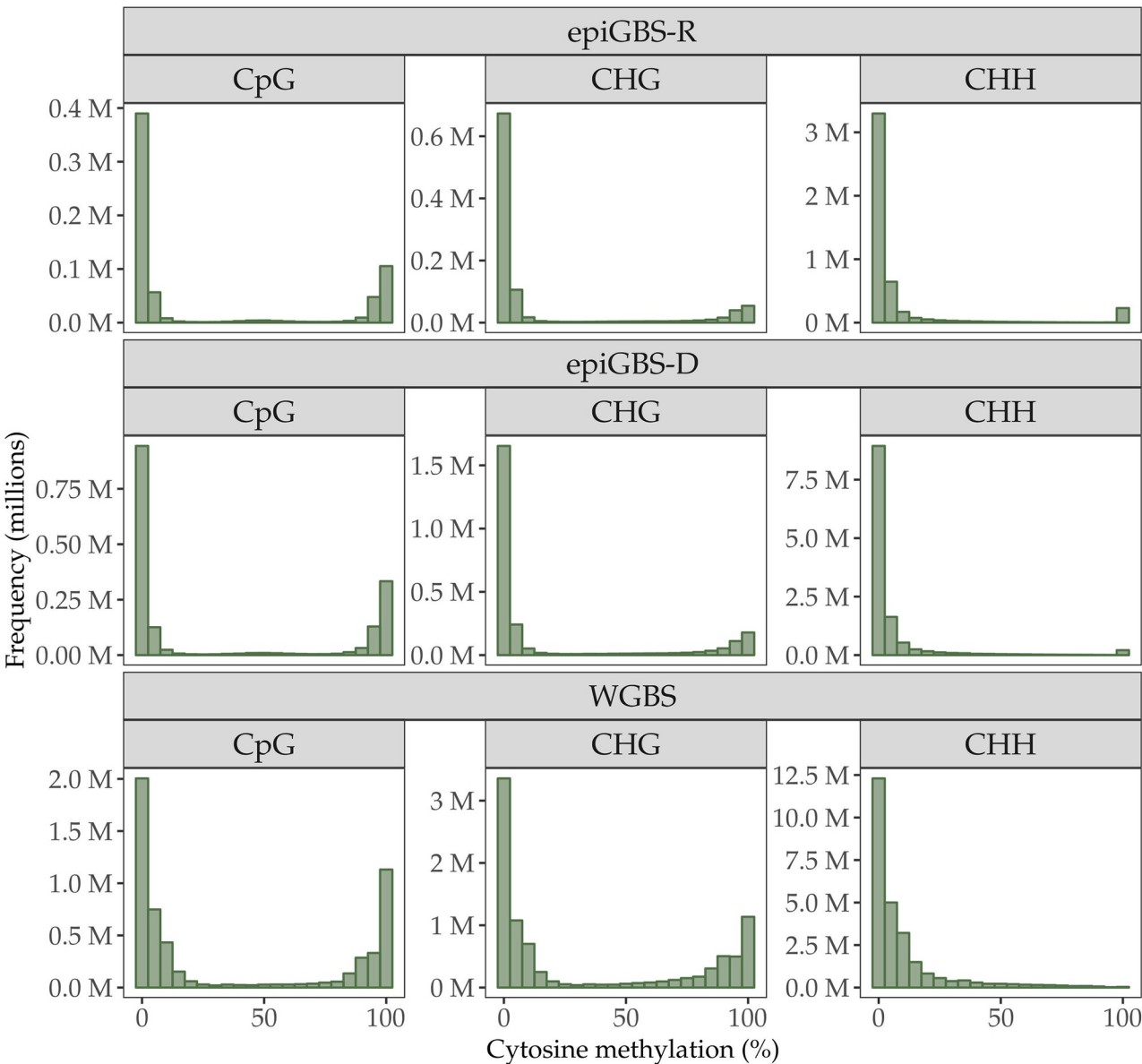

**Fig 1. Histogram of DNA methylation percentage of cytosines captured by epiGBS-R, epiGBS-D and WGBS in CpG, CHG and CHH contexts.**
Methylation percentage was calculated as the mean value across all samples of per-cytosine methylation level.

requirements associated to the large number of CHH positions detected by WGBS (see Table 1). Still, the number of DMCs captured in the CpG and CHG contexts was similar in the two techniques (Pearson's Chi-square test: $X^2$ = 0.02, df = 1, $P$ = 0.88 for CpG context; $X^2$ = 2.12, df = 1, $P$ = 0.15 for CHG context). Furthermore, WGBS captured more DMCs in the CpG context than in the CHG context, whereas a similar number of DMCs were obtained in the two contexts with epiGBS (S1 Table). Overall, the number of DMCs identified in response to artificial herbivory was always greater than those induced by insect herbivory (2794 *vs* 1952 for epiGBS-R, and 5961 *vs* 4714 for WGBS, respectively, in artificial and insect herbivory).

**Table 2. Effect of ortet, herbivory treatment, and their interaction (Ortet × Herbivory) on genome wide average methylation percentage obtained by epiGBS-R, epiGBS-D and WGBS techniques for each of the three sequence contexts (CpG, CHG, and CHH).**

| Technique | Sequence context | Ortet | | Herbivory | | Ortet × Herbivory | |
| --- | --- | --- | --- | --- | --- | --- | --- |
| | | $*F_{2, 18}$ | $P$ | $F_{2, 18}$ | $P$ | $F_{4, 18}$ | $P$ |
| epiGBS-R | CpG | **69.36** | **<0.0001** | 1.50 | 0.25 | 0.56 | 0.69 |
| | CHG | **3.35** | **0.05** | 1.99 | 0.16 | 0.49 | 0.75 |
| | CHH | 0.47 | 0.63 | **7.90** | **0.003** | 1.22 | 0.34 |
| epiGBS-D | CpG | **70.80** | **<0.0001** | 1.26 | 0.30 | 0.63 | 0.64 |
| | CHG | **4.72** | **0.02** | 2.00 | 0.16 | 0.56 | 0.69 |
| | CHH | 0.38 | 0.68 | **9.56** | **0.001** | 1.46 | 0.26 |
| WGBS | CpG | 0.38 | 0.69 | 0.50 | 0.62 | 0.82 | 0.53 |
| | CHG | 0.55 | 0.59 | 0.23 | 0.8 | 0.72 | 0.59 |
| | CHH | 0.37 | 0.7 | 0.08 | 0.92 | 0.65 | 0.64 |

*$F$, degrees of freedom (subscript for $F$) and $P$ values are provided. Values are in bold when $P \leq 0.05$.

Using the WGBS data, we identified a total of 1,057 DMRs, of which 500 DMRs (CpG: 6, CHG: 29; CHH: 465 DMRs) were obtained for insect herbivory and 557 DMRs (CpG: 9, CHG: 47; CHH 501) for artificial herbivory, and none was shared by the two treatments.

## Signs of methylation shifts after herbivory and stress specificity

In CpG context, the proportion of cytosines that shifted to a significantly lower methylation (hypo-methylated DMCs) and those that shifted to a significantly higher methylation (hyper-methylated DMCs) in artificial herbivory was similar as detected either by epiGBS-R and WGBS (Pearson's Chi-square test: $X^2 = 1.3$, df = 1, $P = 0.24$), indicating a higher frequency of hypo-methylation (Fig 3A). For the insect herbivory comparison, the relative frequency of hypo-methylated and hyper-DMCs was rather similar, regardless of the technique (Fig 3A). In CHG context, for both herbivory treatments the relative frequencies of hypo and hyper methylated loci varied depending on the applied technique, slightly more hyper-methylated DMCs were obtained by epiGBS and more hypo-methylated DMCs by WGBS and there appear to be significant differences between DMC numbers ($P < 0.05$; Fig 3A). Finally, in CHH context, a significantly higher number of hypermethylated DMCs was observed in response to insect and artificial herbivory treatments in epiGBS-R (Fig 3A; $P < 0.05$). As mentioned above, WGBS technique was largely inefficient for capturing DMCs in CHH context using the standard DMC calling parameters.

For either insect or artificial herbivory, treatment-specific DMCs (i.e., DMCs that only appear in one of the two herbivory treatments) were by far more abundant than non-specific ones (i.e., DMCs that appear in the two herbivory treatments) in all analyzed contexts, and this trend was found in both epiGBS-R and WGBS (Fig 3B). In particular, the relative frequency of stress specific DMCs was higher in WGBS (92.5% for CpG and 95.5% for CHG) than in epiGBS-R (71.1% for CpG, 75.6% for CHG, 90.1% for CHH).

## Structural annotation of DMCs and DMRs induced by herbivory

The overall results of structural annotation analyses of epiGBS-R data showed that DMCs induced by each of the two herbivory treatments in CpG and CHG contexts were present in all the distinct genome features distinguished. Furthermore, when presence of TEs that

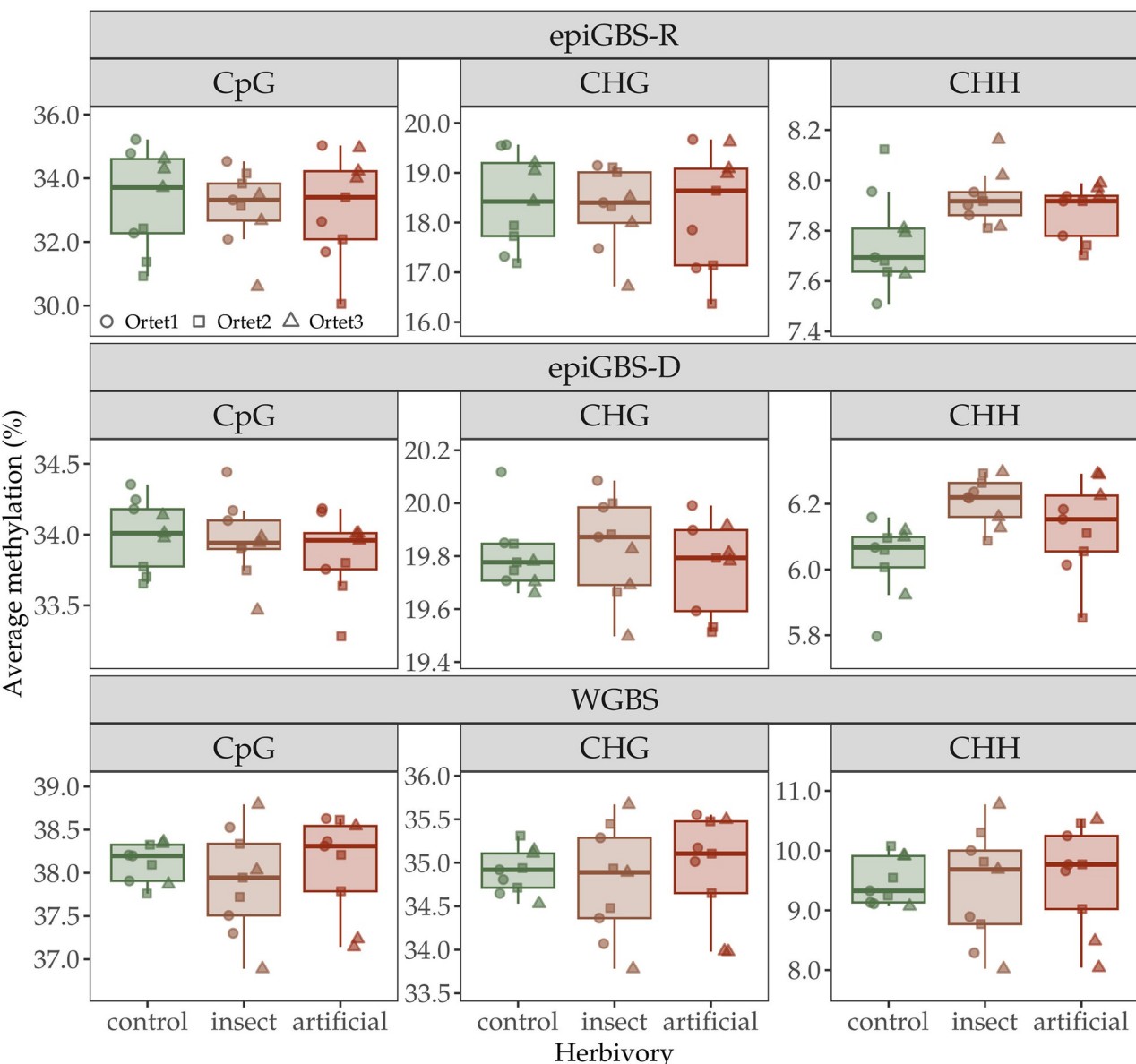

**Fig 2. Average methylation levels (%) captured by epiGBS-R, epiGBS-D and WGBS in the three sequence contexts (CpG, CHG and CHH) in leaves of *P. nigra* cv. italica after insect and artificial herbivory treatment and in non-damaged controls.** The boxplots depict the median, and interquartile range, and whiskers that extend 1.5 times the lower and upper quartiles of n = 9 replicates. Each dot denotes a replicate (ramet), and different symbols were used for the three ortets.

overlapped with the genomic features were indicated, we predominantly found DMCs within the gene body and the intergenic regions which were not overlapping with TEs (Fig 4). However, in the CHH context, much more DMCs were found in the intergenic regions and particularly in those overlapping with TEs. In WGBS data, a similar pattern was observed for CpG and CHG contexts and DMCs were predominantly found in gene bodies not overlapping TEs. As previously stated, WGBS failed to detect DMCs and, thus, no genomic feature was annotated in CHH context for this technique. In general, the amount of DMCs overlapping with TEs detected by WGBS in most genomic features were lower than obtained in epiGBS-R (Fig 4).

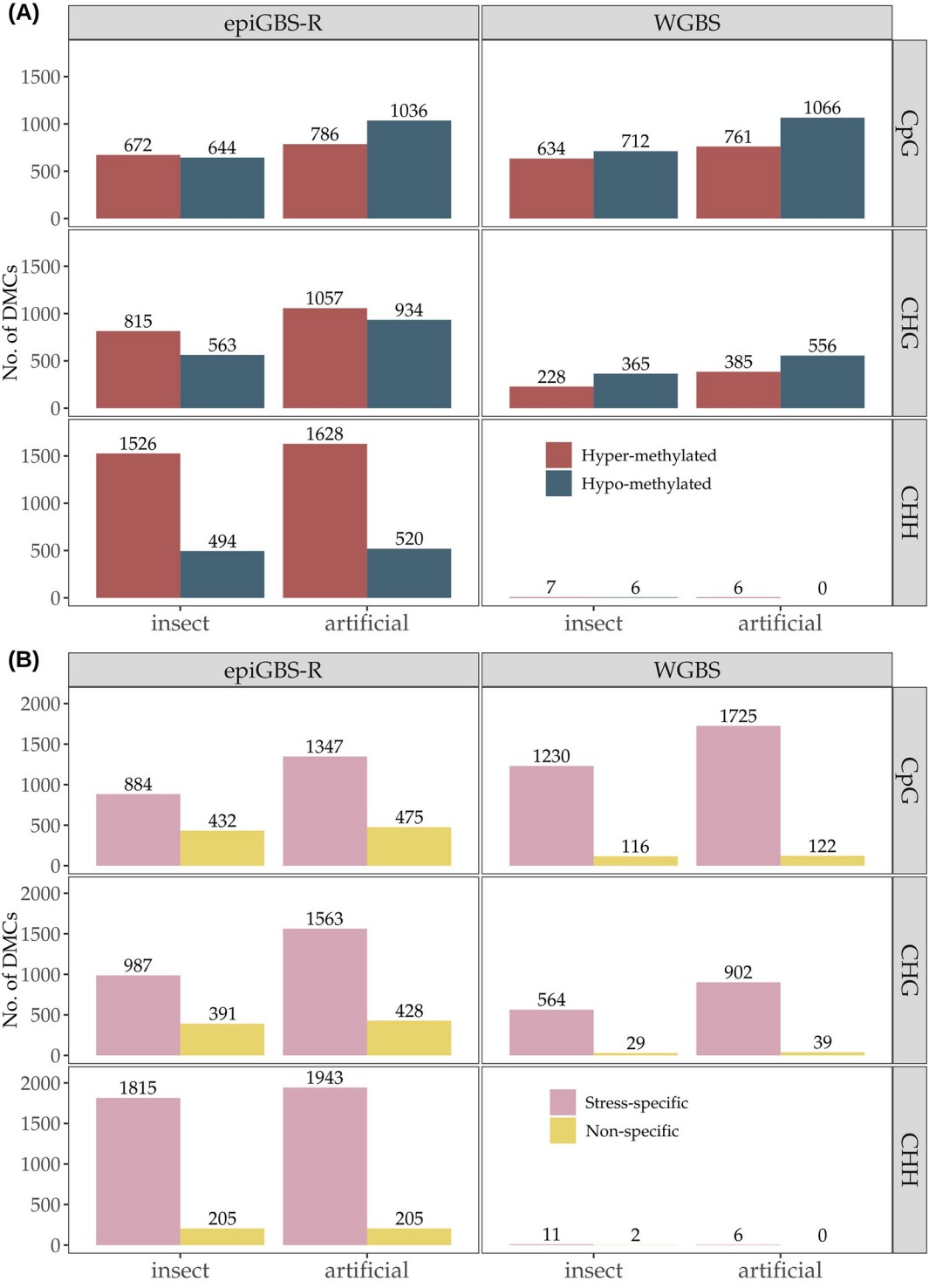

**Fig 3. Differentially methylated cytosines (DMCs) in response to insect and artificial herbivory captured by epiGBS-R and WGBS in the three sequence contexts.** DMCs were defined by a minimum coverage of 10x for epiGBS and 6x for WGBS, 10% change in methylation percentage and q-value <0.05. (A) Sign of methylation shifts after herbivory relative to the methylation status in controls, represented as hyper-methylated DMCs (in red) or hypo-methylated DMCs (in blue). (B) Specificity of the response to either insect or artificial herbivory (stress specific DMCs, in pink; and non-specific DMCs, in yellow).

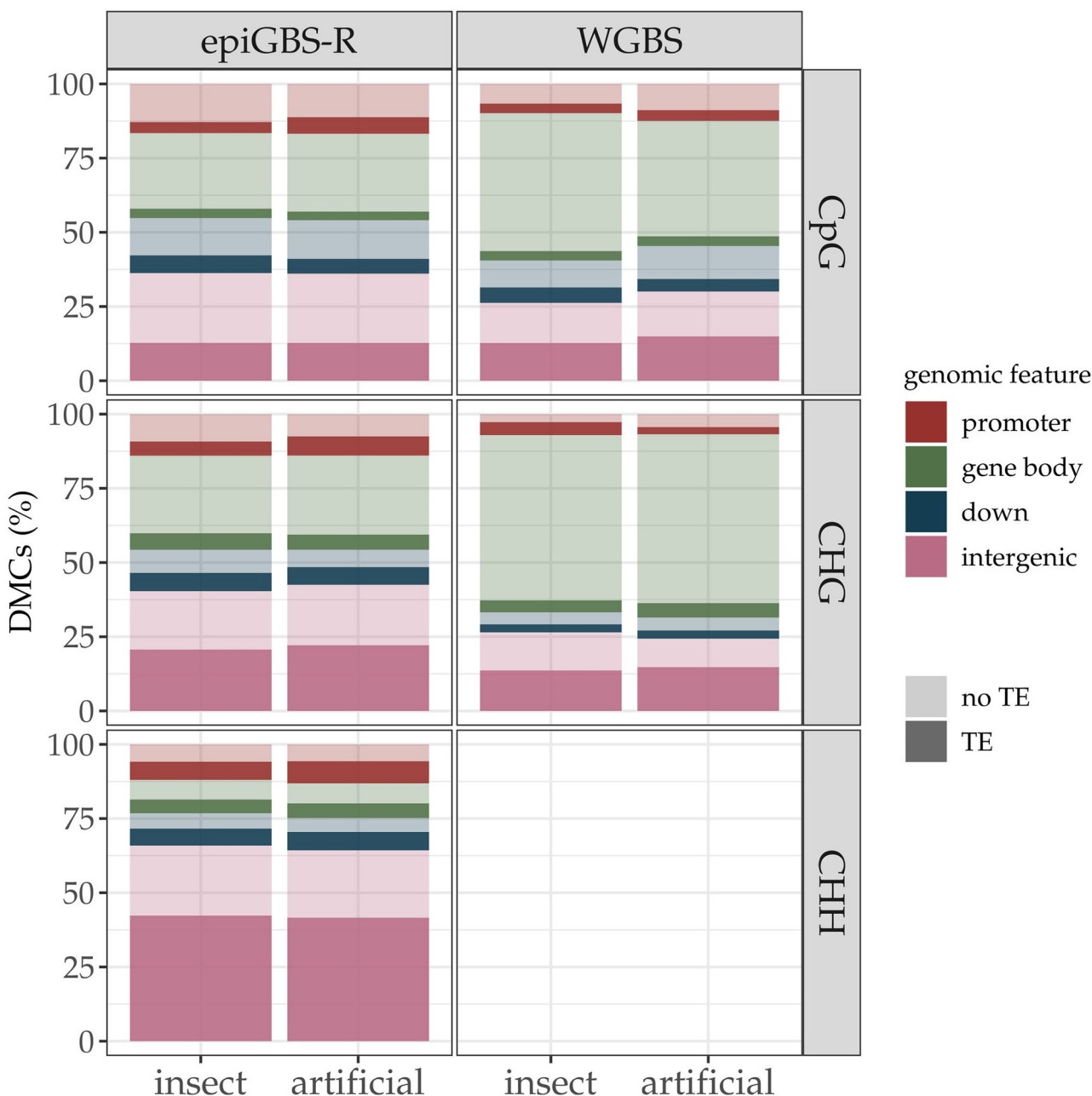

**Fig 4. Genomic location of DMCs detected by epiGBS-R and WGBS in the three sequence contexts.** DMCs mapped within genomic features were classified as being within the gene body (green), promoter (red), and 2 kb down-stream of termination sites (blue) regions. DMCs located out of those regions were assigned to intergenic regions (purple). Information on Transposable Elements (TE) annotation is also indicated, where darker colors indicate TE and lighter colors denote no TE overlapping.

### Signs of methylation shifts after herbivory in the different genomic regions

In both CpG and CHG contexts, and in all the different genomic features, the two techniques captured similar proportions of cytosines that shifted towards lower or higher methylation in response to herbivory (S2 Table). Specifically, in gene bodies not associated with TEs, a higher proportion of hypo-methylated DMCs were detected by epiGBS and WGBS in the two contexts, especially after artificial herbivory (Fig 5A). Also, the two techniques revealed that

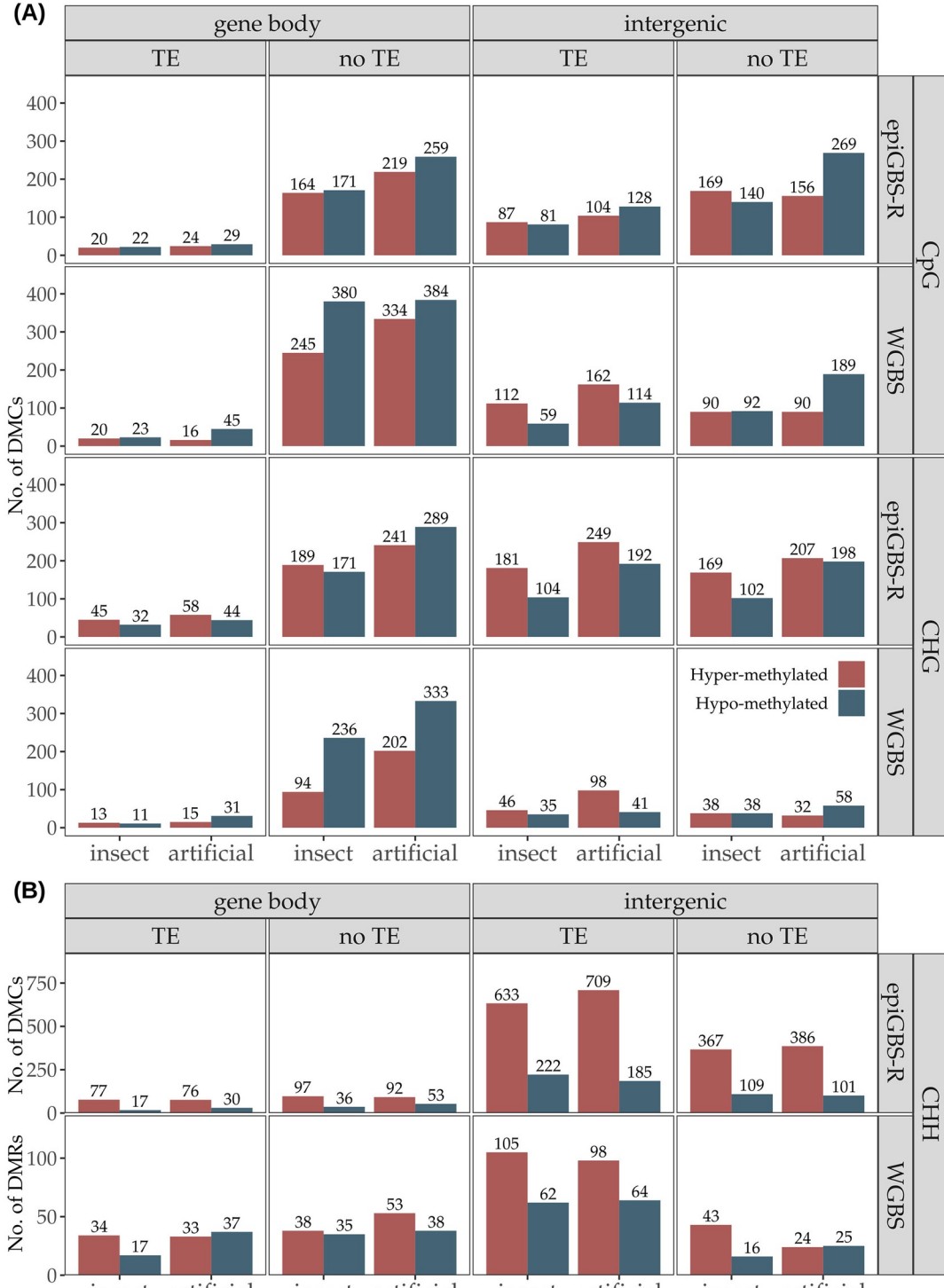

**Fig 5. Methylation shifts in response to insect and artificial herbivory (relative to the methylation status in controls) captured by epiGBS-R and WGBS in the three different contexts that were located in gene bodies and in intergenic regions and their association with TE and non-TE regions.** Hyper-methylated DMCs or DMRs were represented in red, and hypo-methylated in blue. (A) DMCs obtained for CpG and CHG contexts. (B) In CHH, bars represent DMCs obtained for epiGBS-R and DMRs for WGBS. DMCs/DMRs had at least a 10% difference in methylation compared to controls and q-value < 0.05.

DMCs tended to be more hypo-methylated after artificial herbivory in intergenic regions that were not overlapping with TEs in CpG context. Finally, in CHG, the two herbivory treatments tend to produce more hyper-methylation in intergenic regions both in TE and no TE, as detected mainly by epiGBS (Fig 5A).

In CHH context, DMCs captured by epiGBS in all genic features were predominantly hyper-methylated regardless of the type of herbivory experienced, being highest at intergenic regions in both TEs and non-TEs (Fig 5B). Consistent with these results, the largest number of DMRs detected by WGBS in response to the two herbivory treatments were also located at intergenic regions and they were predominantly hyper-methylated (Fig 5B), suggesting that the two techniques were able to detect similar biological response when properly analyzed.

### Functional association of differential methylation changes observed in response to herbivory

None of the GO terms associated to DMCs or DMRs identified by epiGBS and WGBS, respectively, showed enrichment after multiple testing correction (q-value < 0.05), however GO terms selected with uncorrected p-values < 0.05 suggested a functional association. In epiGBS, GO terms with uncorrected $p < 0.05$ indicated that insect herbivory-DMCs enriched genes in biological processes (S2A Fig) related to organelle organization (GO:0006996), phosphorylation (GO:0016310) and some other metabolic processes, while artificial herbivory-DMCs enriched genes in biological processes related to cellular response to stress (GO:0033554), and negative regulation to biological process (GO:0048519).

In WGBS, insect herbivory-DMRs enriched genes in biological processes (S2B Fig) related to defense response (GO:0006952), response to external biotic stimulus (GO:0043207) and defense response to other organism (GO:0098542), while artificial herbivory-DMRs enriched genes in biological processes (S2B Fig) related to response to abscisic acid (GO:0009737) and abscisic acid-activated signaling pathway (GO:0009738). Altogether, using epiGBS DMCs, neither of the treatments enriched GO terms in biological processes similar to WGBS. The full list of enriched gene sets from the DMC and DMR analyses is shown in S3 Table.

### Discussion

In this study, we analyzed DNA methylation in the genomes of Lombardy poplar clones that were either consumed by larvae of the gypsy moth (insect herbivory), hand-defoliated together with application of JA (artificial herbivory) or remained undamaged (control) growing under common conditions. We selected a clonal plant to obtain genetically uniform replicates that could facilitate the identification of independent epigenetic variation, *sensu* [54]. Two single-base resolution methodologies were applied and their outputs compared, the WGBS approach which covered methylation in the entire genome and epiGBS which uses restriction enzymes to reduce the costs of sequencing by interrogating methylation in a portion of the genome [23]. In the following paragraphs, we discuss the extent to which the genome-wide herbivory-induced methylation changes in undamaged leaves grown after treatment were comparable between the two types of herbivory assayed and how epiGBS and WGBS outputs detected these changes in terms of global methylation and differentially methylated loci.

### Overall DNA methylation changes induced by herbivory

Our WGBS analysis provided methylation information on > 116 million cytosines whereas epiGBS was able to analyze between 1.8 and 1.1 million cytosines depending on the use or not of the reference genome available (i.e., for epiGBS-R and epiGBS-D, respectively). In order to understand DNA methylation changes after herbivory, we paid attention to

sequence context as cytosine methylation is introduced and maintained by different methyl-transferase and demethylase systems in the CpG, CHG and CHH contexts of DNA, that might respond differently to external stimuli [2, 3]. We found that DNA methylation was higher in CpG, intermediate in CHG, and lower in CHH, regardless of the technique used for estimation, with estimates at CHG being lower according to epiGBS. Furthermore, all techniques showed a bimodal distribution of cytosine methylation in the CpG context, with either unmethylated or methylated at very high levels, whereas the frequency of highly methylated positions was lower in the CHG and almost absent in CHH, similar to what is typically found in *Arabidopsis* [55].

Variation in average methylation percentages after either artificial or insect herbivory estimated by epiGBS-R and epiGBS-D indicated a significant increased methylation in the CHH context, which is also the context more responsive to short-term stress in other plant species [56, 57]. Methylation percentage in CpG and CHG did not change in response to herbivory but significant variation between the study ortets was observed in these two contexts, likely reflecting intrapopulation variance in methylomes of poplar trees retained after grafting [39, 58]. Methylation estimates based on WGBS did not significantly differ between herbivory treatments or ortets suggesting that the changes associated to the study factors, likely occurring in a small fraction of positions, were not reflected as an overall global methylation change with this technique. A higher sequencing depth in WGBS could partially circumvent the limitation observed here [59–61]. Summing up, in our study epiGBS seemed to be efficient for identifying the global methylation changes associated to herbivory, suggesting that the genome sampling accomplished was representative to detect so, although differential methylation analyses should be more informative (see below).

## Differential methylation analyses: Stress and context specific changes

The main purpose of applying single base resolution bisulfite sequencing methods is to capture differential methylation loci and their location according to genomic features that could be instrumental to understanding responses to a certain factor [62, 63]. These analyses require subsequent data filtering because reliable estimates of methylation for single evaluated positions need to be represented in most if not all samples (e.g., we required that every analyzed position was present in at least six out nine samples per group). In our filtered dataset, epiGBS and WGBS obtained comparable numbers of DMCs in CpG and CHG contexts for a change in methylation between control and treated plants larger than ten percent. In these two sequence contexts, all methods revealed that most captured DMCs were specific to one of the two herbivory treatments assayed. This result is in line with the limited overlap between the transcriptome response to similar experimental treatments already reported in poplar [36], but contrasts somehow with the shared methylation response in CpG and CHG contexts induced by pathogen infection and salicylic acid exposure in *A. thaliana* [57]. Artificial herbivory produced a higher number of DMCs than insect herbivory. Looking at the three sequence contexts we found that in CpG context, the proportion of cytosines that shifted to a significantly lower methylation (hypo-methylated DMCs) and those that shifted to a significantly higher methylation (hyper-methylated DMCs) were similar after insect herbivory whereas a higher number of hypo-methylated DMC were observed after artificial herbivory. In CHG context the ratio of hyper and hypo-methylated DMCs was not captured analogously by different methods, artificial and insect herbivory tended to show more hyper-methylated DMCs according to epiGBS and the opposite trend was obtained in WGBS. Previous studies showed that in rice, heavy metal treatment induced hypo-methylation in CHG, whereas in *Arabidopsis*, hypo- and hyper-methylation modifications in CHG context were produced by an avirulent strain or the

defense hormone salicylic acid [57, 64] indicating that both hyper- and hypo-methylation can be obtained in response to certain stress factors.

Interestingly, in CHH context, epiGBS found the highest number of DMCs, with a clear prevalence of hyper-methylated DMCs, whereas WGBS obtained only a negligible number of DMCs, likely due to the reduced statistical power (because of multiple testing correction) to detect changes in methylation at positions that usually have very low methylation level when the number of evaluated positions is large. This pattern was observed in our study where 90,235,706 CHH sites were present in WGBS compared to the 877,942 in epiGBS raw data, but a minor percentage was retained by WGBS when only positions properly covered in all samples were selected (Table 1). For this reason, in WGBS, detection of differentially methylated regions (DMRs), *i.e.* contiguous stretches of DNA sequence in the genome that show differing levels of DNA methylation between groups of samples, has become a much frequent approach, provided that variation in DMR has been also associated with phenotypic plant variation (e.g., [59, 65]). When we applied the standard DMR identification method for WGBS, we found that they were most frequently found in CHH context and hyper-methylated DMRs were more frequent after either insect or artificial herbivory. Thus, in CHH context, the DMR output of WGBS showed similar relative frequency and sign than obtained by DMCs in epiGBS. A response characterised by hyper-methylated DMRs suggested that de novo methylation in CHH islands would be a suitable response associated to herbivory in the Lombardy poplar (see e.g., [66]). This is an interesting finding and may have implications for gene transcription as CHH hyper-methylation in the proximity of a gene reduced the level of transcription of that particular gene in the apple tree [58].

Overall, the epiGBS technique, which can be applied to any species in the absence of a reference genome at a lower cost per sample, was successful in demonstrating that methylation changes induced by insect and artificial herbivory occurred at distinct loci and indicated that increased methylation in the CHH context was the most frequently observed response. Thus, epiGBS could be particularly useful for non-model plant species and large experimental designs, such as those intended to search for species-specific epigenetic responses in plants typically damaged by a diverse array of antagonistic animals or pathogens, or the impact of multiple levels of abiotic conditions (e.g., temperature, water availability, and their combination) that could better simulate different scenarios of climate change.

## Structural annotation and functional association of DMCs and DMRs induced by herbivory

Despite the fact that epiGBS usually covers a small percentage of the study genome (in our case 1.5%), our study found no apparent bias towards association of DMCs to specific genomic feature compared to those obtained by WGBS after artificial and insect herbivory in Lombardy poplar clones, supporting that the method provided a sound genome-wise analysis [27]. In particular, DMCs detected in CpG and CHG contexts were more frequently associated to hypo-methylation in gene bodies regardless of the method applied, an interesting finding provided that in *P. trichochocarpa* methylation in gene bodies had a more repressive effect on transcription than methylation in promoters [67] and therefore the observed hypo-methylation might be associated with the upregulation of genes transcription (see also [36]). Conversely, the most frequent change observed in CHH was hyper-methylation in TEs and intergenic regions. Such findings are consistent with response after pathogen infection in which increased CHH methylation levels associated to TEs were found in *A. thaliana* [57] and the response in the offspring of *Mimulus* plants exposed to leaf damage [21]. Additionally, hyper-methylation was suggested to be a general defense mechanism against pathogen stress in tobacco plants [68].

As a further step to better understand the potential consequences of methylation changes observed, we conducted gene enrichment analyses associated to DMRs from WGBS and DMCs from epiGBS to compare their outputs, provided there is currently no unified method for performing enrichment analyses of DMRs from both approaches. Gene enrichment analysis from WGBS-DMR revealed that most changes associated with herbivory were related to the Gene Ontology (GO) category associated to biological processes, but enrichment in specific GO categories differed between insect herbivory and artificial herbivory. Insect herbivory was more related to responses to biotic stimuli, defense responses and immune system processes, whereas changes associated with artificial herbivory were more related to abscisic acid stimulus, ligase and cell cycle process, and mRNA processing. The genes associated to epiGBS-DMCs were equally associated to biological process, molecular function and cellular components categories, mainly related to catabolic processes or organelle organization within the cell. Therefore, epiGBS is useful for understanding the localization and direction of differential methylation, but it does not directly reveal specific functional response in genes associated to DMCs, a constraint that is an inherent limitation of epiGBS (and RRBS in general) likely because the majority of genes in the genome are not included in the fraction of the genome analyzed.

## Conclusion

To sum up, we found that epiGBS offered reliable insight about methylation changes in DNA of Lombardy poplar clones experiencing insect and artificial herbivory. The results offered by epiGBS and WGBS were consistent as regards (i) the context dependent response, mainly associated to increased methylation in CHH, (ii) the specificity of the response elicited by insect and artificial herbivory, indicated by the few shared DMCs, and (iii) the structural annotation of those changes, mainly associated to TEs and intergenic regions for CHH, and to gene bodies and their flanking regions in CpG and CHG. Thus, epiGBS succeeded to characterize global, genome-wide methylation changes in response to a certain stress, being particularly useful for investigating species lacking a reference genome, whereas WGBS performed better in the functional analysis. The functional interpretation (at the level of GO term enrichment) of the observed methylation changes remained unclear and additional transcriptome analyses might be instrumental to characterize the epigenetic regulation of stress-specific responses in non-model plants with limited genomic resources.

## Supporting information

**S1 Fig. Mean coverage distribution for each position across all samples obtained with epiGBS and WGBS.**
(TIF)

**S2 Fig. Gene ontology (GO) clustering and selected significantly enriched GO terms of DMCs and DMR-associated genes.**
(TIF)

**S1 Table. Total number of differentially methylated cytosines for insect and artificial treated plants captured by WGBS and epiGBS-R (reference branch) in the three cytosine contexts.**
(DOCX)

**S2 Table. Total number of differentially methylated cytosines (DMC) captured by epiGBS-R (reference branch) and WGBS in the three sequence contexts (CpG, CHG and CHH) in each genomic feature (promoter, down-stream of transcriptional sites, gene body**

**and intergenic region), within transposable elements (TE) or outside (no TE) also indicated.**
(DOCX)

**S3 Table. Full list of enriched gene sets from the DMC and DMR analyses.**
(XLSX)

**S1 Appendix. Technical analysis of global methylation with fragments shared by epiGBS-R and WGBS.**
(PDF)

## Acknowledgments

We thank Slavica Milanovic-Ivanovic for her technical assistance in the molecular laboratory during the creation of the epiGBS and WGBS libraries, respectively. We thank Fleur Gawehns' guidelines for using the epiGBS pipeline and troubleshooting, and Morgane van Antro and M. Teresa Boquete for insightful discussions. We thank Bhumika Dubay's contribution to the reference genome, the gene model, and the TE predictions. We also thank Bárbara Díez-Rodríguez for providing us with poplar cuttings and Sybille Unsicker for the gypsy moth caterpillars. We appreciate all of the contributions and conversations with the members of the Epidiverse Consortium, and the suggestions provided by Susmita Barman and an anonymous referee.

## Author Contributions

**Conceptualization:** A. Niloya Troyee, Cristian Peña-Ponton, Mónica Medrano, Koen J. F. Verhoeven, Conchita Alonso.

**Data curation:** A. Niloya Troyee, Cristian Peña-Ponton.

**Formal analysis:** A. Niloya Troyee, Cristian Peña-Ponton.

**Funding acquisition:** Koen J. F. Verhoeven, Conchita Alonso.

**Investigation:** A. Niloya Troyee, Cristian Peña-Ponton.

**Methodology:** A. Niloya Troyee, Cristian Peña-Ponton, Conchita Alonso.

**Project administration:** Koen J. F. Verhoeven, Conchita Alonso.

**Resources:** Koen J. F. Verhoeven.

**Supervision:** Mónica Medrano, Koen J. F. Verhoeven, Conchita Alonso.

**Visualization:** Cristian Peña-Ponton, Mónica Medrano.

**Writing – original draft:** A. Niloya Troyee, Mónica Medrano, Conchita Alonso.

**Writing – review & editing:** Cristian Peña-Ponton, Mónica Medrano, Koen J. F. Verhoeven, Conchita Alonso.

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
