## [Decision Letter · Decision Letter 0]

26 Jul 2023

PONE-D-23-09704Herbivory induced methylation changes in the Lombardy poplar: a comparison of results obtained by epiGBS and WGBSPLOS ONE

Dear Dr. Alonso,

Thank you for submitting your manuscript to PLOS ONE. After careful consideration, we feel that it has merit but does not fully meet PLOS ONE’s publication criteria as it currently stands. Therefore, we invite you to submit a revised version of the manuscript that addresses the points raised during the review process.

We look forward to receiving your revised manuscript.

Kind regards,

Samikshan Dutta, Ph.D.

Academic Editor

PLOS ONE

4. We notice that your supplementary figures are uploaded with the file type 'Figure'. Please amend the file type to 'Supporting Information'. Please ensure that each Supporting Information file has a legend listed in the manuscript after the references list.

Reviewers' comments:

Reviewer's Responses to Questions

**Comments to the Author**

1. Is the manuscript technically sound, and do the data support the conclusions?

Reviewer #1: Partly

Reviewer #2: Partly

2. Has the statistical analysis been performed appropriately and rigorously? 

Reviewer #1: Yes

Reviewer #2: No

3. Have the authors made all data underlying the findings in their manuscript fully available?

Reviewer #1: Yes

Reviewer #2: Yes

4. Is the manuscript presented in an intelligible fashion and written in standard English?

Reviewer #1: Yes

Reviewer #2: No

5. Review Comments to the Author

Reviewer #1: Both EpiGBS and WGBS utilize bisulfite treatment and sequencing to study DNA methylation, EpiGBS is a targeted approach focusing on specific genomic regions, whereas WGBS provides a comprehensive analysis of the entire genome. The choice between these techniques depends on the research goals, available resources, and the desired level of coverage and resolution.

The study is good and well designed. I have few questions and comments:

Comment:

Poor formatting, figure legends look misplaced. The legends look like the part of results and have no sense where they are placed. It makes this article difficult to understand.

Questions:

Fig S1

WGBS covers the whole genome which contains repetitive region while epiGBS covers specific genomic regions, so the results are expected.

Table 1

Name NA is confusing, do the authors have any specific reason to use name with/without NA? I am not able to connect it with study design.

In Table 1 epiGBS-R/D captured most cytosines in CHH condition while WGBS has least %age cytosine capture in CHH. What is your view on this?

Table 2

In epiGBS-R/D, ortet shows significant methylation at CpG, herbivory show significant methylation at CHH. Ortet and herbivory interaction has no significant methylation. What is the biological meaning of this result?

Table S1

Why does WGBS have less DMCs than epiGBS-R (line 326)?

Overall, the study says that epiGBS offers reliable methylation insight. But the major drawback is that WGBS is performing better than epiGBS in functional analysis, so how epiGBS can be useful for actual biological system?

Reviewer #2: In their article titled "Herbivory induced methylation changes in the Lombardy poplar: a comparison of results obtained by epiGBS and WGBS," Troyee et al. intended to present data from epiGBS and WGBS by assessing the global and sequence-specific differential methylation induced by herbivory.

This constitutes a large, exhaustive, and rationally broad body of work that is appreciable. Listed below are a number of major concerns that can be addressed to enhance the content of the manuscript.

1. Please revise the Abstract to specify the study's objective, findings, and significance.

2. Please include an abbreviation list detailing all abbreviations used in the manuscript. As there are a few terms that are not adequately explained in the manuscript.

3. Although the methylation change is more pronounced in CHG and CHH, please also discuss the methylation change in CH due to the various herbivores listed in Tables 1,2), as well as the CG changes, which will demonstrate the insignificance of the CH methylation profile axis.

4. The entire manuscript must be rewritten because the arrangement is not clear. Like why is it necessary to compare epiGBS and WGBS? What is the purpose of the research? Please focus on rewriting these elements.

5. The grammatical construction requires additional attention.

6. Please include a PRIZMA chart for the data search.

6. PLOS authors have the option to publish the peer review history of their article (what does this mean?). If published, this will include your full peer review and any attached files.

Reviewer #1: No

Reviewer #2: **Yes: **Susmita Barman

---

## [Author Response · Author response to Decision Letter 0]

8 Aug 2023

Reviewer #1: I have few questions and comments:

Q1. Poor formatting, figure legends look misplaced. The legends look like the part of results and have no sense where they are placed. It makes this article difficult to understand.

R1. According to the Plos One formatting guidelines “Each figure caption should appear directly after the paragraph in which they are first cited” and “Figures should be uploaded separately as individual files.” We followed this rule and Figure captions appear as required, following the paragraph in which the figure is first quoted. The edited version should look different.

R1. In addition, in order to improve the value of captions in Figures and Tables we have made an effort in shortening and sharpening captions of Fig 2, Table 1 and Table 2.

Q2. Fig S1. WGBS covers the whole genome which contains repetitive region while epiGBS covers specific genomic regions, so the results are expected.

R2. We were uncertain how to use this comment beause the coverage reached by a sequencing experiment is not always equally successful. We decided to keep the figure as supporting information and have shortened the sentence to better clarify the result and improve readability. [L222] “…data from the two sequenced libraries indicated that epiGBS got higher coverage for the captured cytosines (S1 Fig).” The sentence was moved to Methods section (see response to the next comment)

Q3. Table 1. Name NA is confusing, do the authors have any specific reason to use name with/without NA? I am not able to connect it with study design.

R3. Information in Table 1 is related to filtering criteria and shows the number of cytosine positions captured by each method depending on the sample representation specified. We have moved it to Methods to better indicate the meaning of what we previously mentioned as NA (=Not Available) or missing data. The full explanation now reads: [L221] “We applied different minimum read coverage because data from the two sequenced libraries indicated that epiGBS got higher coverage than WGBS for the captured cytosines (S1 Fig). Individual databases were later merged using the unite function of MethylKit that kept bases covered by 2/3 of the samples per treatment group (i.e., in six out of nine samples). We retained a total of 1,823,024 (epiGBS-R), 1,148,755 (epiGBS-D) and 116,785,713 (WGBS) cytosines. Further, in order to compare only positions with methylation calls that were common to all samples, a dataset was built without any missing values (i.e., data available in 100% of samples). The number of cytosine positions in the three sequence contexts captured by each technique after read coverage filtering and the two sample representation options are shown in Table 1.”

We have also modified the Table 1 and its Caption indicating that “Sample representation” explains the difference between columns data. The new caption reads “Table 1. Total number of cytosines captured by epiGBS-R, epiGBS-D and WGBS in the three sequence contexts (CpG, CHG and CHH) according to the sample representation threshold applied.” And all the other details are indicated as a footnote “*Values indicate the number of cytosines included in each dataset after doing the minimum read coverage filtering (≥5 sequencing coverage in WGBS; ≥10 in epiGBS-R and epiGBS-D), taking into account their presence in at least 2/3 of study samples per group or being common to all study samples.”

After this remark, we have thought the output of the libraries could be better placed at Methods too. We have moved that information to the end of the respective “library construction, sequencing and pre-processing” section [L194 and L212]. Altogether, the content of section “Output of the epiGBS and WGBS libraries” has moved to Methods section in the revised version.

Q4. In Table 1 epiGBS-R/D captured most cytosines in CHH condition while WGBS has least %age cytosine capture in CHH. What is your view on this?

R4. Firstly, it should be noted that the differences between contexts are minor compared to the differences obtained for the comparison between the three techniques. WGBS was the technique that screened the greater number of cytosines compared to all epiGBS options, regardless of whether we considered cytosines present in all samples or six out of nine per group. 

 And secondly, if we should speculate about the specific reasons for a lower percentage of CHH positions common to all samples obtained in WGBS, we would say that CHH may have low read coverage resulting in missing data in a higher number cases when both cytosine positions (N = 90,235,706 cytosines) and samples (N = 27 ramets) screened. We have added a comment in this line at Results [L335] “likely due to reduced statistical power associated to multiple-testing correction requirements associated to the large number of CHH positions detected by WGBS (see Table 1)”. Also, we improved explanation at Discussion [L471] “This pattern was observed in our study where 90,235,706 CHH sites were present in WGBS compared to the 877,942 in epiGBS data, but a minor percentage was retained by WGBS when only positions properly covered in all samples were selected (Table 1)”

Q5. Table 2. In epiGBS-R/D, ortet shows significant methylation at CpG, herbivory show significant methylation at CHH. Ortet and herbivory interaction has no significant methylation. What is the biological meaning of this result?

R5. This result is in line with previous studies showing context-specific responses as mentioned in the discussion [L558-559] “cytosine methylation is introduced and maintained by different methyl-transferase and demethylase systems in the CpG, CHG and CHH contexts of DNA, that might respond differently to external stimuli [2,3]” and [L564-568] “Variation in average methylation percentages after either artificial or insect herbivory estimated by epiGBS-R and epiGBS-D indicated a significant increased methylation in the CHH context, which is also the context more responsive to short-term stress in other plant species [1,2]. Methylation percentage in CpG and CHG did not change in response to herbivory but significant variation between the study ortets was observed in these two contexts, likely reflecting intrapopulation variance in methylomes of poplar trees retained after grafting [3,4]”

From a technical viewpoint, the statistical power to detect a significant interaction might be lower, particularly if the expected variation for a certain factor (e.g., ortet) would be in the magnitude and not in the sign of methylation shift induced by the other factor (e.g., herbivory). 

Q6. Table S1. Why does WGBS have less DMCs than epiGBS-R (line 326)?

R6. WGBS having less DMCs than epiGBS-R was mainly due to a lack of captured DMCs at CHH context, likely because of low methylation in most positions in CHH context and multiple testing correction. We have explained this at Discussion [L469] “in CHH context, […] WGBS obtained only a negligible number of DMCs, likely due to the reduced statistical power (because of multiple testing correction) to detect changes in methylation at positions that usually have very low methylation level when the number of evaluated positions is large [57-59].”

It is also relevant to consider the fact that [L478] “in CHH context, the DMR output of WGBS showed similar relative frequency and sign than obtained by DMCs in epiGBS.”

Q7. Overall, the study says that epiGBS offers reliable methylation insight. But the major drawback is that WGBS is performing better than epiGBS in functional analysis, so how epiGBS can be useful for actual biological system?

R7. EpiGBS enables researchers to examine genome-wide DNA methylation patterns and changes in a large number of samples, and identify biologically significant epigenetic variation even in absence of a reference genome, providing sufficient information to answer questions based on study objectives and available economic resources. To address this question, we have added a comment in Discussion [L483] “Overall, the epiGBS technique, which can be applied to any species in the absence of a reference genome at a lower cost per sample, was successful in demonstrating that methylation changes induced by insect and artificial herbivory occurred at distinct loci and indicated that increased methylation in the CHH context was the most frequently observed response. Thus, epiGBS could be particularly useful for non-model plant species and large experimental designs, such as those intended to search for species-specific epigenetic responses in plants typically damaged by a diverse array of antagonistic animals or pathogens, or the impact of multiple levels of abiotic conditions (e.g., temperature, water availability, and their combination) that could better simulate different scenarios of climate change.”

Reviewer #2: Listed below are a number of major concerns that can be addressed to enhance the content of the manuscript.

Q1. Please revise the Abstract to specify the study's objective, findings, and significance.

R1. We appreciate the suggestion. We have modified the abstract to be more specific as regards the objective: “In this study, we provide for the first time a comprehensive comparison between the outputs of RRBS and WGBS when applied to characterize DNA methylation changes in response to a single environmental factor.” 

We also modified the first sentence related to results to be more explicit: “We found that, after any of the two herbivory treatments, global methylation percentage increased in CHH, and the shift was detected as statistically significant only by epiGBS.”

Finally, we closed the abstract adding a sentence to highlight the significance of the results: “Our results support that epiGBS could be particularly useful in large experimental designs aimed to explore epigenetic changes of non-model plant species in response to multiple environmental factors.”

Q2. Please include an abbreviation list detailing all abbreviations used in the manuscript. As there are a few terms that are not adequately explained in the manuscript.

R2. We have a made a careful revision of all abbreviations used along the MS. We feel most of the acronyms are commonly used in the field but we agree they can be unknown for the general audience of PLOS One. We have scrutinized the text to review the explanation provided for all of them. In order to clarify the meaning of the acronyms CpG, CHG, CHH we have now specified H = Adenine, Thymine, Cytosine both in abstract [L16] and the first time they are mentioned in main text [L39]. 

For the acronyms related to the study techniques, we have made sure that upper case was used to correctly indicate their respective meaning: Whole Genome Bisulfite Sequencing (WGBS) and Reduced Representation Bisulfite Sequencing (RRBS), both in abstract and the first time they are mentioned in main text [L57]. The same was done with Jasmonic Acid (JA) [L110], and Transposable Elements (TE) [L284]. The acronyms for Differentially Methylated Cytosines (DMCs) and Differentially Methylated Regions (DMRs) were correctly indicated within the titles of their respective sections.

Apart from that, we deleted some unnecessary acronyms that were not used more than twice (GBS, VOC, NNN, SA, NA, TSS). Altogether, we think that after the changes addressed, the abbreviation list is not required but we will be eager to include it as a supporting information upon editor’s request.

Q3. Although the methylation change is more pronounced in CHG and CHH, please also discuss the methylation change in CH due to the various herbivores listed in Tables 1,2), as well as the CG changes, which will demonstrate the insignificance of the CH methylation profile axis.

R3. Sorry, unfortunately we were unable to understand exactly what the reviewer want to say in this comment (e.g., no reference to treatment appeared in Table 1), so we could not give any reply at this point or change the manuscript accordingly.

Q4. The entire manuscript must be rewritten because the arrangement is not clear. Like why is it necessary to compare epiGBS and WGBS? What is the purpose of the research? Please focus on rewriting these elements.

R4. We appreciated the comment and focused in improving the main issue highlighted, as we agreed it is a very interesting item. In order to clarify the purpose of the research we have included two new sentences at Introduction to explain the motivation of the comparison conducted based on:

(i) [L63] The quality of the [WGBS] output will vary with genome features (e.g., genome size, frequency of repeats), sequencing depth reached and the quality of the annotated reference genome, which limits its application to non-model plants with unknown genome features [23].

(ii) [L77] If the two techniques identify similar global and context-specific methylation shifts and point to similar genomic location of most of the observed methylation changes in response to a certain level of environmental stress, the epiGBS analyses could be useful to explore the links between epigenetic variation and plant functional phenotypic traits in non-model plants, with typical ecological designs involving large sample sizes and multiple levels of environmental variation that have mainly used anonymous markers [15, 16, 19] or indirect evidence of epigenetic contribution to stress response [13].

Q5. The grammatical construction requires additional attention.

R5. We appreciate the advice although we missed some more specific suggestions. We have done a careful revision all throughout the manuscript and edit sentences that were too long, could be rewritten in active form, or contain some error. We indicate here just some examples. [L159] “We collected tissue from undamaged and fully expanded leaves of the adjacent apical half of each ramet, either 24 hours after the second herbivory event in treated plants or after the aqueous spraying in controls. We kept these leaves without any bag cover throughout the duration of the experiment.” And [L368] “The overall results of structural annotation analyses of epiGBS-R data showed that DMCs induced by each of the two herbivory treatments in CpG and CHG contexts were present in all the distinct genome features.”

Q6. Please include a PRIZMA chart for the data search. 

R6. We were unable to understand this suggestion. The PRISMA chart is a valuable tool for literature search and study selection process in meta-analysis. However, we have conducted an experiment and generated data ourselves. We did not conduct a systematic literature review of the use of the two techniques and, thus, we cannot provide the info required for referring such study.

---

## [Editor Report · Decision Letter 1]

24 Aug 2023

Herbivory induced methylation changes in the Lombardy poplar: a comparison of results obtained by epiGBS and WGBS

PONE-D-23-09704R1

Dear Dr. Alonso,

We’re pleased to inform you that your manuscript has been judged scientifically suitable for publication and will be formally accepted for publication once it meets all outstanding technical requirements.

Kind regards,

Samikshan Dutta, Ph.D.

Academic Editor

PLOS ONE

---

## [Editor Report · Acceptance letter]

29 Aug 2023

PONE-D-23-09704R1 

Herbivory induced methylation changes in the Lombardy poplar: A comparison of results obtained by epiGBS and WGBS 

Dear Dr. Alonso:

I'm pleased to inform you that your manuscript has been deemed suitable for publication in PLOS ONE. Congratulations! Your manuscript is now with our production department. 

Kind regards, 

on behalf of

Dr. Samikshan Dutta 

Academic Editor

PLOS ONE